# Effects of Monascus on Proteolysis, Lipolysis, and Volatile Compounds of Camembert-Type Cheese during Ripening

**DOI:** 10.3390/foods11111662

**Published:** 2022-06-06

**Authors:** Shuwen Zhang, Tong Wang, Yumeng Zhang, Bo Song, Xiaoyang Pang, Jiaping Lv

**Affiliations:** Institute of Food Science and Technology, Chinese Academy of Agricultural Sciences, Beijing 100193, China; zhangshuwen@caas.cn (S.Z.); 15811499853@163.com (T.W.); zym18811735237@163.com (Y.Z.); songbocaas@163.com (B.S.); pangxiaoyang@caas.cn (X.P.)

**Keywords:** monascus-ripened cheese, proteolysis, lipolysis, flavor

## Abstract

In order to improve the flavor and taste of Camembert cheese, the use of Monascus as an adjunct starter for the production of Camembert-type cheese was studied to investigate its effect on the proteolysis, lipolysis, and volatile compounds during ripening for 40 days. The Camembert cheese without Monascus was used as a control. The results showed that proteolytic and lipolytic activities increased to a certain extent. The addition of Monascus promoted primary and secondary proteolysis, due to the release of some proteases by Monascus. Aspartic, Threonine, Glutamic, Glycine, Methione, Isoleucine, Phenyalanine, and Lysine contents in experimental group (R) cheese were significantly higher than those in control group (W) cheeses. In addition, the free amino acid and fatty acid contents were also affected. The identification of flavor components using gas-mass spectrometry (GC-MS) showed that 2-undecone, 2-tridecanone, phenylethyl alcohol, butanediol (responsible for the production of flowery and honey-like aroma), ethyl hexanoate, ethyl octanoate, and ethyl citrate (fruit-like aroma) were significantly higher (*p* < 0.05) in the experimental cheeses than in the control. The contents of 2-nonanone, 2-octanone and 2-decanone (showing milky flavor), and 1-octene-3 alcohol with typical mushroom-like flavor were lower than the control.

## 1. Introduction

Camembert cheese is a typical surface-ripened mold cheese with a white bloomy rind made from raw milk in regions of Camembert (Normandy), France. Camembert-type cheese ripens quickly due to a high moisture content and rapid growth of surface mold. During ripening, a series of enzymatic reactions and chemical changes occurred in the surface-ripened cheeses which had important effects on their texture and flavor. Presently, *Penicillium camembert* and *Geotrichum candidum* are the main fungi starters used in the production of Camembert cheese, and it can be purchased through commercial channels [1]. Several researchers have recently carried out some works for the selection of suitable starters for improving their sensory qualities and nutrition. The Korean traditional red ginseng powder was used for the production of Camembert type cheese in order to improve its functional characteristics [2]. In addition, three strains selected from 129 yeast species with potential probiotic functions were used for the production of Camembert cheese [3]. 

Monascus has been used in food for thousands of years, mainly in traditional applications such as fermented bean curd, red koji rice, red wine, and red koji vinegar. In recent years, Monascus-fermented foods have become increasingly popular worldwide. Monascus was used as an adjunct starter in the making of semi-hard cheese [4]. Monascus fermented products are popular because of its natural red color and unique flavor. Many studies have shown that the Monascus fermented products contained several bioactive substances, such as Monacolin K, (GABA), Monascus pigments, polysaccharides, etc. [5]. Monascus metabolites GAGB and Monacolin K can be used for the treatment and prevention of hypertension, the reduction of cholesterol, and the improvement of circulatory system, respectively [6,7]. Therefore, the application of Monascus as an adjunct starter for improving the flavor and quality of Camembert type cheese would be greatly significant in the development of new cheeses product, especially in China.

In this study, Camembert-type cheeses were produced using Monascus as an adjunct starter. During 40 days for ripening, the proteolysis and lipolysis as well as the inherent volatile compounds of the experimental cheese were analyzed and compared with the control. The aim of this study was to analyze the effect of Monascus on the physicochemical composition and flavor substances during the ripening of Camembert type cheese.

This study is expected to provide scientific basis and technical guidance for the production and quality control of red mold cheese during ripening.

## 2. Materials and Methods

### 2.1. Materials

Raw milk was purchased from Sanyuan Lvhe Farm, Beijing, China. Monascus grains were bought from China center of industrial culture collection (CICC, No. 5038, Beijing, China). *Penicillium camembert* (CHOOZIT TM PC 12 LYO 20D Danisco) was purchased from Danisco (Gournier, France), while lactic acid bacteria starter (FD-DVS R-704 PHAGE CONTROL) and rennet (CHY-MAX POWDER EXTRA NB) were obtained from Chr. Hansen (Hessholm, Denmark). Analytical grade chemicals used in this study were purchased from Sinopharm Chemical Reagent (Shanghai, China), while chromatographic grade chemicals were purchased from Thermo Fisher Scientific (Pittsburgh, PA, USA).

### 2.2. Cheese Manufacture and Sampling

The Camembert-type cheeses were produced from whole cow milk with or without the addition of Monascus. The experimental groups (R cheese) were produced with a combination of commercial starter and Monascus, while the control groups (W cheese) were produced only with the conventional commercial starter. Monascus fermentation broth was first prepared by inoculating strains of Monascus into PDA medium for 3–4 days (30 °C). The spores formed were collected by diluting with sterile water to obtain 1.0 × 10^5^ cfu/mL spores solution. Secondly, the spores solution were inoculated into sterilized 5% whole potato powder solution with shaking culture for 7 d (30 °C, 200 rpm). Fresh whole milk (60 L) was pasteurized (75 °C for 15 s) in a pasteurizer (Powerpoint Corporation, Tokyo, Japan) and then cooled to 35 °C. The pasteurized milk was divided into two vats of 30 L each. The commercial starter containing *Lactococcus lactis subsp. lactis*, *Lactococcus lactis subsp. cremoris* (0.015 g/L) and *Penicillium camembert* (2.1 × 10^4^ spores per milliliter) were inoculated in each tank. At the same time, the 5% Monascus fermentation broth was added into one tank and gently stirred for 5 min to obtain an even mixture. The other tank of milk was used to make the control cheeses. After 110 min, when the pH of milk drops to 6.5, the rennet (0.02 g/L) was added. The milk was allowed to coagulate for 40 min, and the curds were cut into square shapes (1 cm^3^) after hardening. The mixtures of curd and whey were left standing for 10 min in the tank. Then, the curds were loaded into plastic molds with 105 mm diameter and 30 mm height. The average weight of cheeses was 220 ± 20 g. After 4 h, the cheeses were smeared with 1% (*w*/*w*) dry salt and then transferred to a sterilized ripening chamber, where they were stored for 24 h under a controlled condition (14 °C and 70% relative humidity (RH)). Ripening conditions were changed for an additional 14 d (14 °C and 90% RH). Lastly, at day 16, the cheeses were wrapped with wax paper and further ripened at 4 °C until day 40.

### 2.3. Proteolysis

Proteolysis was determined according to the method of Chen et al. (2013), based on the value of total nitrogen (TN), pH of 4.6, acid-soluble nitrogen (ASN), and 12% trichloroacetic acid (TCA) soluble nitrogen (NPN). Each analysis was done in triplicate.

For the determination of free amino acids, 5 g of cheese samples were dissolved in 20 mL of 5% (*m*/*v*) trichloroacetic acid solution with an ultraturrax homogenizer (IKA, Germany) at 10,000× *g* for 2 min. The slurry was centrifuged (High Speed Refrigerated Centrifuge, Sigma, Neustadt, Germany) for 10 min at 10,000× *g* and 4 °C, and then filtered through a double-layer filter paper. The supernatant was passed through an 0.22 μm filter, and then analyzed in an Agilent HPLC coupled with AB5500 QQQ mass spectrometer. All analyses were conducted in triplicate.

### 2.4. Lipolysis

The acid degree value (ADV) was determined according to the method described by Katsiari et al. [8]. The method of Chavarri et al. was applied for the determination of free fatty acids [9]. All analyses were conducted in triplicate.

### 2.5. Volatile Compounds

The samples were prepared according to the method of Lee et al. (2018) with some modifications [2]. Each sample (5.0 g) was transferred into a 20 mL glass vial with a teflon-lined septum and then immediately sealed with an aluminum seal. The samples were agitated and melted for 30 min at 60 °C in order to accelerate the equilibrium volatile compounds. Then, the volatile compounds were extracted using a 50 μm Carboxenpoly-coated SPME fibre (Shanghai ANPEL Scientific Instrument Co., Ltd., Shanghai, China). The fibres obtained were exposed to the glass headspace for 30 min prior to adsorption. The analyses of volatile compounds were performed on a gas Chromatogram coupled with a mass spectrometer (GC-MS) (QP2010Plus, Shimadzu, Kyoto, Japan). Volatile compounds were separated using a capillary column (DB-5 ms; 30 m × 250 μm ID × 0.25 μm film thickness; Agilent, Santa Clara, CA, USA). The carrier gas was helium with a flow rate of 0.8 mL/min. The oven temperature was programmed initially at 40 °C for 3 min, later raised to 130 °C (6 °C/min, held 3 min), and finally raised to 230 °C (8 °C/min). The conditions of mass spectrometry conditions were set as follows: electronic impact mode at 70 eV; emission current of 200 A; interface temperature at 250 °C; source temperature at 200 °C; and test voltage of 350 V. Electron impact (EI) mass spectra were identified using the NIST (version 2.0) mass spectral library, and the mass spectra were compared with those contained in the NIST. The volatile compounds were quantitatively determined based on the calculated ratio from the internal standard calibrations of tetrahydrofuran (890 mg/g).

### 2.6. Statistical Analysis

All the data obtained were expressed as means values of triplicate independent determinations. The results were expressed as mean values ± standard deviations (SD). The significance of differences between the data was assessed using one-way analysis of variance (ANOVA) by Dunnett’s tests, where * *p* < 0.05, ** *p* < 0.01, *** *p* < 0.001.

## 3. Results and Discussion

### 3.1. Effects of Monascus on Proteolysis and FAA of Camembert-Type Cheese

Proteolysis plays an important role in the texture and flavor of cheeses during ripening; therefore, the degree of proteolysis is considered as an important indicator of cheese maturation. In general, the index proteolysis is usually expressed as the level of pH, acid soluble nitrogen (ASN/TN%), and non-protein nitrogen (NPN/TN%). Proteolysis in Camembert-type cheeses is mainly catalyzed by protease from starters, residual rennet, and indigenous milk proteinases. ASN/TN is an index of primary proteolysis. The production of ASN is mainly about proteolysis catalyzed by residual rennet in cheeses, and were partly generated by proteases from microorganisms. NPN/TN is an indicator of secondary proteolysis, and the content of NPN is related to protease origin from mold and bacteria in cheeses [10]. Figure 1 showed that the values of ASN/TN and NPN/TN increased significantly in both samples with the ripening time, which suggested that primary and secondary protein hydrolysis were increasing. The rate of proteolysis in the later stage was lower than that in the primary stage. In the previous period of ripening time (0–16 d), microbes, especially mold, grew rapidly and produced a large amount of proteases to accelerate proteolysis. Then, cheeses were wrapped and placed at 4 °C after 16 d, and the lowered environmental temperature produced an adverse effect on the microbial growth and protease activity. Figure 1 also showed that the addition of Monascus promoted primary and secondary proteolysis, due to the release of some proteases by Monascus. The increasing of proteolysis could also be attributed to the Monascus fermented solution containing some nutritive materials capable of promoting the growth of fungi and bacteria [4].

The concentration of individual free amino acids (FAAs) in cheeses was affected by the processing, type, and degree of proteolysis, ripening time and conditions, etc. [11]. FAA content in both cheeses increased obviously as the ripening time (Figure 2) and correlated with the results of the proteolysis due to the fact that the addition of Monascus promoted the hydrolysis of protein in Camembert-type cheeses. Aspartic, Threonine, Glutamic, Glycine, Methione, Isoleucine, Phenyalanine, and Lysine contents in R cheese were significantly higher than those in W cheeses. FAAs contribute to the flavor of cheeses through two pathways. On the one hand, some of the FAAs are tasteful substances. According to their different tastes, FAAs can be divided into five types, I class (acid fresh): Glu, Asp; II class (sweet fresh): Thr, Ser, Gly, Met and Cys; III class (sweet slightly bitter): Pro, Ala; IV class (bitter): Val, Leu, Ile, Phe, and Tyr; V class (bitter slightly sweet): His, Lys, and Arg. The content of the first four classes in R cheeses was higher than that in W cheeses, which indicated that the addition of Monascus strengthens the taste of umami and sweetness of the cheese. On the other hand, FAAs serve as a precursor substance of the catabolic reaction that generates a number of flavor compounds, such as aldehydes, alcohols, carboxylic acids, thiols, etc. There are three main kinds of amino acids as the precursors of flavor synthesis. Aromatic amino acids (Trp, Phe, Tyr) were converted into benzaldehyde, phenyl alcohol, and other substances under the catalyzing of aromatic amino invertase; branched chain amino acids (Val, Leu, Ile) under the catalyzing of branched amino invertase or aromatic amino invertase, are transformed into methyl acid, methyl aldehyde, isobutyl ester, etc.; methionine (Met) convert into some volatile sulphides under the action of enzymes [12]. The three kinds of amino acids in R cheeses were significantly higher than those in W cheeses, which illustrated that the addition of Monascus promoted the production amino acids and provided more sufficient substrates for the synthesis of flavor substances to improve the flavor of Camembert-type cheese.

### 3.2. Effects of Monascus on Lipolysis and FFA of Camembert-Type Cheese

The degree of lipolysis of cheese is generally determined by acid degree value (ADV), which reflects the total amount of free fatty acids in cheeses. In Figure 3, the ADV value of both cheeses followed similar trends during ripening. At the previous stage of ripening (0 to 24 d), the ADV initially increased rapidly and later slowly increased because the lipases were released from molds in Camembert cheeses [13]. The temperature and relative humidity at the initial stage (0–16 d) was higher, resulting in fast growth of molds, and a large amount of lipases were produced to catalyze the hydrolysis of triglyceride. After 24 d, the decrease of temperature and relative humidity led to the decrease of the activity of lipases. In addition, the fatty acid was gradually consumed as a precursor; thus, the growth rates of ADV at a later stage were slowed down. The ADVs of the R cheeses were higher than those of the W cheeses, probably due to the acceleration of lipolysis by Monascus in cheeses. Some studies found that the content of unsaturated fatty acids was highly metabolized by Monascus, which may consequently enhance the hydrolysis of triglycerides by Monacolins [4,5].

Free fatty acids (FFAs) play a very important role in the ripening of Camembert-type cheese. Free fatty acids are not only volatile flavor compounds but also important precursors for the synthesis of alcohols, ketones, and esters [14]. The lengths of detected carbon chain in the fatty acids are 4 to 18. The FFAs can be divided into three groups according to their numbers of carbon atoms: short-chain (4:0–8:0), medium-chain (9:0–13:0), and long-chain (14:0–C18:0). Among them, the short-chain fatty acids have strong volatility with some pungent odors and are easily used as substrates in enzyme-catalyzed reactions to synthesize other flavoring substances. As the carbon atoms increased, their activities decreased. FFAs with carbon number more than 12 have little volatility and have little effect on sensory quality [15]. The content of FFAs increased during 0–16 d, then the growth rate of most fatty acids decreased after 16 d. (Figure 4). There was no significant difference in the content of FFAs between both cheeses in 0 d. As the ripening time evolves, the contents of C4:0, C6:0, and C12:0 in R cheeses were significantly higher than that of W cheeses while the content C8:0, C10:0 was lower than that of W cheeses. From the result of ADV, we know that the addition of Monascus promotes the hydrolysis of triglycerides. On the other hand, the fatty acid is consumed as a flavor precursor to produce other flavor compounds during the ripening. The different rates of utilization of various FFAs in cheeses may be accountable for the differences among FFAs [16]. Some esterases released during the metabolism of Monascus may also influence the utilization of FFAs. In addition, some long-chain unsaturated FFAs were detected in both cheeses and showed no obvious fluctuation during ripening. However, the contents of unsaturated FFAs in R cheeses were higher than those of W cheeses. Some studies revealed that most unsaturated FFAs derived from the metabolism of Monascus, which may explain the reason why there are more unsaturated fatty acids in R cheeses.

### 3.3. Effects of Monascus on Volatile Compounds of Camembert-Type Cheese

GC-MS analysis was performed to identify the volatile compounds isolated from both Camembert-type cheeses during ripening. A total of 45 compounds were detected in W cheeses; which was comprised of 11 acids, 11 ketones, 10 alcohols, 8 esters, and 6 unclassified compounds. On the other hand, 47 compounds were detected in R cheeses which included 11 acids and 10 ketones, 10 alcohols, and 8 esters and 7 unclassified compounds. As the ripening time extended, the four types of flavor substances showed the same trend in both cheeses, but there were differences in the percentage and concentration. Similar results were previously reported [17,18].

#### 3.3.1. Acids

Acid compounds in Camembert-type cheese originated from three main biochemical pathways during ripening time; which include lipolysis, proteolysis, and lactose fermentation. Linear fatty acids (carbon atoms > 4) originate from lipolysis, while branched acids were mainly obtained from the deamination of amino acids from proteolysis [3]. Statistically significant differences (*p* < 0.05) during the maturation of both cheeses were identified in most acids (Table 1). The carboxylic acids derived from the hydrolysis of triglyceride (butyric acid, caproic acid, heptanoic acid, caprylic acid, citric acid, decanoic acid, myristic acid, and lauric acid) were the most abundant, accounting for more than 90% of the total content of acids. This suggests that acids were mainly derived from lipolysis in W cheeses and R cheeses. Carboxylic acids are not only aroma compounds, but also precursors to produce other important flavor compounds such as methyl ketones, alcohols, lactones, aldehydes and esters [19]. As shown in Figure 5, the content of acids increased continuously from 0–16 d, and gradually began to decrease after 16 d. As a result of the rapid growth of molds at the earlier stages (0–16 d), a lot of lipolytic enzymes were released which accelerated the rate of lipolysis in cheeses. Therefore, the fatty acid content also increased rapidly. After 16 d, lipolytic activity was hindered by the decrease in ripening temperature (from 14 °C to 4 °C) and some FFAs were consumed as precursors. When the synthetic rate of FFAs was lowered than the conversion rate of formation, there was a decreasing trend. Short-chain FFAs and medium-chain FFAs are considered the key factors of flavor which share the similar tendency of total concentration of acids. Among them, caproic acid, butyric acid, methyl butyric acid, and caprylic acid are known for the most effective odorants in Camembert-type cheeses. Caproic acid has a spicy and sour taste; methyl butyric acid and butyric acid possess a typical sweaty and cheesy taste. Caprylic acid is one of the typical flavor components of goat cheeses [20,21]. The contents of butyric acid, caproic acid, and lauric acid in R cheeses were significantly higher than those of W cheeses (*p* < 0.05), while the contents of caprylic acid and citric acid were lower than those of W cheese (Table 1). It was presumed that Monascus and their metabolins affected the growth and metabolism of *Penicillium* as well as the environment of lipolysis. Moreover, some acids were gradually consumed during the synthesis of other flavored substances with ripening time. The utilization of different fatty acids varied. Acetic acid and propionic acids are products of the fermentation process of lactose [22]. Acetic acid and propionic acid initially increased during 0–24 d and then decreased later (Table 1), which indicated that the bacteria responsible for the fermentation were most active during the early stages. Their activity was later inhibited as a result of the decrease of temperature, thus consequently reducing the growth of bacteria as well as their biochemical reactions. W cheese contains significantly higher levels of acetic and propionic acid than R cheese, due to the presence of Monaco in R cheese which inhibits the growth of bacteria and further inhibits their release. Srianta et al. reported that Monascus are capable of producing some bacteriostatic substances [7]. The contents of 2-methylbutyric acid and 3-methylbutyric acid were higher in the R cheeses, probably for the promotion of proteolysis by Monascus observed in our results. However, only a few studies about proteases and lipases from Monascus were reported. Therefore, metabolic pathways of protein and triglyceride in cheeses caused by Monascus needs further investigation.

#### 3.3.2. Ketones

Ketones play key roles in the flavor of the surface-ripening mold cheeses due to their typical odor and low threshold. As shown in Figure 5, the ketones of both cheeses increased with ripening time. The ketones identified in both cheeses are mainly methyl ketones, which are the most abundant neutral compounds in the volatile components of mold-ripened cheeses especially Camembert and Blue Cheeses [23]. Most methyl ketones obtained carbon chains with odd number of carbon atoms (C3~C15), except 2-hydroxybutanone. Among them, 2-pentanone, 2-heptanone, 2-octanone, 2-nonanone, and 2-nonanone were characteristic flavor substances of surface-ripened cheeses [24]. Methyl ketone is synthesized from free fatty acid under the catalysis of some enzymes [25]. The results of GC-MS revealed that the content of methyl ketone had a negative correlation with those of the corresponding precursor (FFAs) (Table 1). The contents of 2-butanone, 2-undecylone, and 2-tridecanone in R cheeses were significantly higher than that in W cheeses (*p* < 0.05). However, the content of 2-nonanone, 2-octanone, and 2-decanone in R cheeses were lower than those of W cheeses (*p* < 0.05) (Table 1). It was presumed that the formation of the product led to the consumption of the precursor material. Therefore, the effect of Monascus on the methyl ketone in cheeses was directly linked to their effect on lipolysis. As shown in Table 1, W cheese has a higher content of octanoic acid and its corresponding methyl ketone (2-anthrone) is higher than that of R cheese, while R cheese has a higher content of lauric acid and its corresponding methyl ketone (2-undecanone) is higher than that of W cheese. The flavors of ketones are diverse, with 2-pentanone, 2-undecanone, and 2-tridecanone providing a typical fruit flavor; 2-anthone, 2-octanone, and 2-hydroxybutanone are the main substances of milk aroma; heptanone is related to the flavor of mushroom; 2-nonanone and 2-decanone provide a baked aroma [13,22]. From the above results, we can know that R cheese contains more ketones with fruit aroma substances while the W cheese contains more ketones with milk aroma.

#### 3.3.3. Alcohols

1-octene-3-ol, phenylethyl alcohol, methyl butanol, and butylene glycol were the main alcohol detected in two cheeses. As the ripening time prolonged, the alcohols in the two cheeses increased rapidly in the previous stage (0–16 d) and gradually decreased after 16 d (Figure 5). In the later period (16–40 d), the reduction rate of alcohols in R cheese was lower than that of W cheese although the alcohol content decreased continually in both cheeses. At the last day of experiment (40 d), the content of alcohol in R cheese was significantly higher than that of W cheeses (*p* < 0.05). With the ripening time, 1-octene-3-ol accumulated continuously in both cheeses and up to the highest at the end of ripening. The principal difference between the two types of cheese was the content of phenyl ethanol. 1-octene-3-ol was the most abundant alcohol in W cheeses, while phenyl ethanol was the most abundant alcohol in R cheese (Table 1). 1-octene-3-ol is considered to be the most typical component of the mushroom flavor of Camembert and Blue Cheese with a low threshold. Phenyl ethanol has a typical floral aroma and honey aroma [26,27]. After one week of ripening, the content of phenylethyl alcohol reached a maximum in W cheese and then gradually decreased. In R cheese, the content of phenylethyl alcohol reached the maximum at 16 d and then just a slight decrease happened. Finally, the content of phenylethyl alcohol in R cheese is significantly higher than that of W cheese (Table 1). 2-heptanol and 2-nonanol (with alcoholic aroma) were the main secondary alcohols identified in both cheeses, which correlated with the variation of their corresponding precursors (methyl ketones). The content of butanediol correlates with the content of their corresponding methyl ketone (2-butanone), which explains the differences between the two cheeses.

#### 3.3.4. Esters

Most esters provide floral and fruity aroma and reduce the spicy and bitter tastes from FFAs and amines. The content of esters in both cheeses increased until 32 d and then became stable. However, a slight decrease was observed in a few samples (Figure 5). As shown in Table 1, the esters identified in both cheeses were ethyl esters and lactone. It was generally recognized that the synthesis of ethyl esters mainly by esterification and alcoholic reactions, under the catalysis of esterase and alcohol acyltransferase, respectively. However, there were other ways to produce ethyl esters, such as acid hydrolysis and transesterification, with the influence of some lactic acid bacteria, yeasts, or molds [28]. The content of esters in cheeses depends on the balances between the synthesis and hydrolysis of esters. It is mainly affected by the environmental reactions (water activity, substrate concentration, pH et al.) which are in a dynamic change. Therefore, the content of esters is in a dynamic equilibrium. There are no detailed reports on how to increase or decrease the hydrolysis of esters. Lactone is a cyclic ester, which was formed by intermolecular esterification of alkyd. α-lactone and β-lactone are intermediate products of organic synthesis. γ-Lactone and σ-lactone are relatively stable and not easily decomposed. Lactone offers a strong aromatic odor [29]. Although these odors are not true cheese flavors, they have a close relationship with the formation of cheese flavor [30]. The content of lactone in the W cheeses was relatively stable, while there was a slight increase in R cheeses during the later ripening period. The ethyl esters gave the cheeses a fruity flavor, which promoted the balance of the overall flavor of the cheeses. Ethyl acetate, ethyl butyrate, ethyl hexanoate, ethyl octanoate, and ethyl decanoate are common in Camembert cheeses and which type and content vary in different types of cheeses [27]. The content of esters in R cheeses were higher than those of W cheeses during ripening. It could be attributed to the fact that R cheese contained more esters as a result of the Monascus fermented broth containing some esters or esterase possibly. In addition, several studies have shown that the esterase released by Monascus promoted the synthesis of esters such as ethyl hexanoate and ethyl octanoate [17,18]. In addition, there were more alcohols in R cheeses according to the result of 3.3.3, which provided more substrates for the synthesis of esters in R cheeses.

#### 3.3.5. Other Compounds

Other compounds, including pentanal, benzaldehyde, d-limonene toluene, butylated hydroxytoluene, decane, octane, and ammonium acetate, were also detected in both cheeses. Aldehydes are temporary compounds that could be rapidly reduced to primary alcohols or oxidized into corresponding acids rather than accumulated in cheese. Benzaldehyde, known for its typical nutty flavor, derived from the conversion of tryptophan and phenylalanine, promoting the overall flavor of the cheeses. d-dimonene is a terpenoid with lemon and orange aroma. The formation of benzene ring compounds such as toluene and butyl hydroxytoluene may be related to the transformation of tryptophan and phenylalanine. Hydrocarbons and acetic salts contributed little to flavor and were present in very low amounts in the cheeses.

In the future, we hope to use new flavor analysis techniques such as non-destructive testing to evaluate flavor change patterns during cheese ripening [31].

## 4. Conclusions

Monascus as an adjunct starter was used in the processing of Camembert cheese. The addition of Monascus promoted the proteolysis and lipolysis of Camembert and affected the content of FAAs and FFAs. The content of methyl ketones, alcohols, and esters made a difference in the change of FAAs and FFAs. Esterase released from Monascus played a vital role in the utilization of acids and alcohols as well as the synthesis of ethyl esters. The gas-mass spectrometry (GC-MS) for the determination of the flavor components showed that 2-undecone, 2-tridecanone, phenylethyl alcohol (responsible for the flower-like, honey-like aroma), ethyl octanoate, and ethyl citrate (responsible for the fruity aroma) were significantly higher in the R cheeses. 2-nonanone, 2-octanone, and 2-decanone (with milky flavor); and 1-octene-3 alcohol (with typical mushroom flavor) present lower contents in R cheese.

## Figures and Tables

**Figure 1 foods-11-01662-f001:**
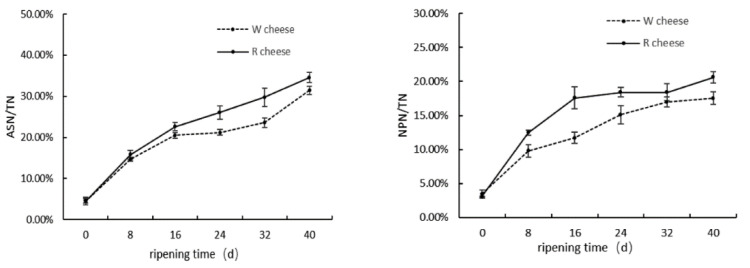
Changes of ASN/TN and NPN/TN of two cheeses with different ripening time.

**Figure 2 foods-11-01662-f002:**
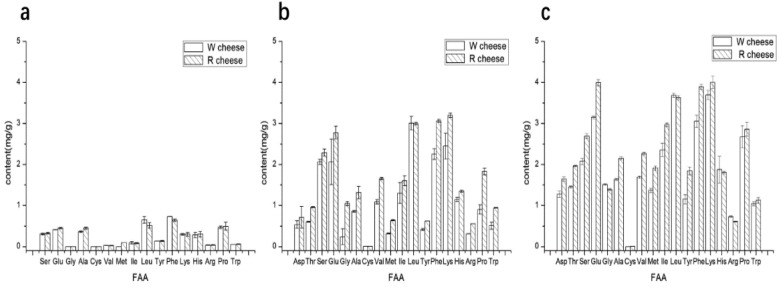
Content of individual free amino acid (FAA) of cheeses with different ripening time. (**a**) 0 d; (**b**) 16 d; (**c**) 40 d.

**Figure 3 foods-11-01662-f003:**
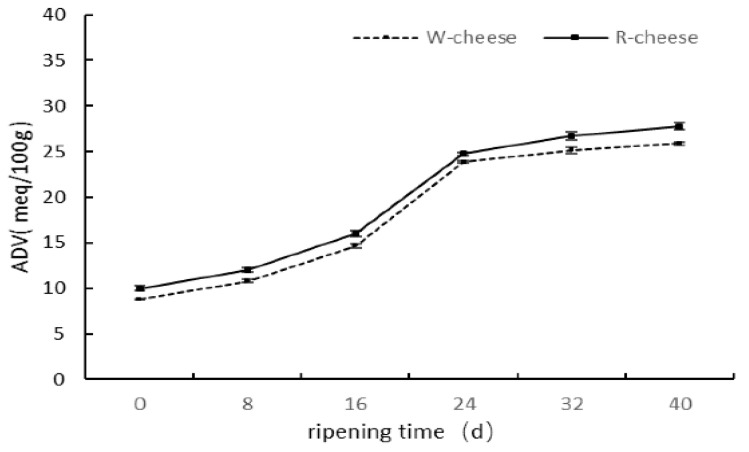
Changes of ADV of two cheeses with different ripening times.

**Figure 4 foods-11-01662-f004:**
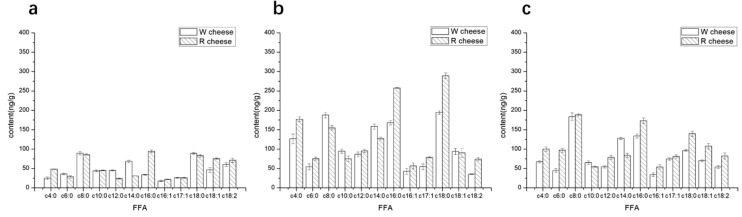
Content of individual free fatty acid (FFA) of cheeses with different ripening time. (**a**) 0 d; (**b**) 16 d; (**c**) 40 d.

**Figure 5 foods-11-01662-f005:**
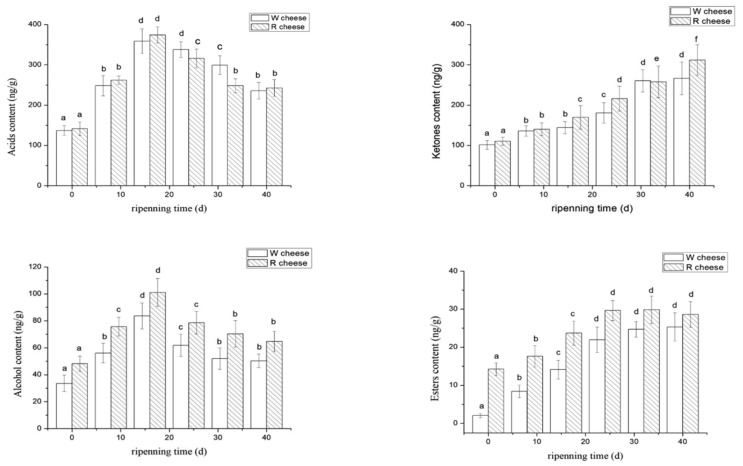
Changes of volatile compounds isolated from two cheeses with different ripening times. The significance of differences not sharing a common letter are significantly different (*p* < 0.05).

**Table 1 foods-11-01662-t001:** Volatile compounds (ng/g) isolated from cheeses with different ripening times.

RI	Compounds	Cheese	Ripening Time (d)					
			0	8	16	24	32	40
	Acids							
594	acetic acid	W	2.82 ± 0.37 ^Aa^	7.42 ± 0.68 ^Ab^	16.27 ± 1.25 ^Ac^	43.15 ± 5.51 ^Ae^	34.18 ± 3.16 ^Ad^	23.68 ± 3.22 ^Ac^
		R	0.73 ± 0.35 ^Ba^	1.49 ± 0.18 ^Bb^	13.96 ± 0.56 ^Bc^	26.92 ± 1.35 ^Be^	19.06 ± 0.18 ^Bd^	13.37 ± 1.65 ^Bc^
661	propionic acid	W	2.27 ± 0.12 ^a^	13.70 ± 0.96 ^Ac^	25.36 ± 0.82 ^Ab^	20.79 ± 1.48 ^Ab^	17.81 ± 1.35 ^Ab^	16.59 ± 2.29 ^Ab^
		R	ND	0.66 ± 0.04 ^Ba^	1.29 ± 0.73 ^Bb^	5.74 ± 0.56 ^Bc^	9.06 ± 1.72 ^Bd^	10.57 ± 2.17 ^Bd^
775	Butyric acid	W	14.31 ± 1.23 ^Ab^	22.08 ± 0.26 ^Bc^	21.40 ± 4.26 ^Bc^	15.25 ± 6.35 ^Bb^	22.90 ± 2.53 ^Ac^	8.68 ± 0.20 ^Ba^
		R	11.27 ± 3.47 ^Ba^	52.69 ± 1.16 ^Ad^	63.38 ± 1.99 ^Ae^	38.86 ± 7.89 ^Ac^	20.51 ± 2.23 ^Ab^	21.59 ± 3.19 ^Ab^
861	2-Methyl butanoic	W	24.81 ± 2.61 ^Aa^	35.48 ± 1.65 ^Bb^	47.08 ± 17.73 ^Bd^	40.74 ± 1.73 ^Ac^	34.64 ± 0.44 ^Ab^	33.32 ± 0.47 ^Bb^
		R	23.90 ± 0.51 ^Aa^	58.79 ± 2.39 ^Ac^	56.42 ± 1.27 ^Ae^	42.98 ± 0.25 ^Ad^	34.83 ± 1.15 ^Ac^	38.29 ± 0.93 ^Ab^
877	3-Methyl pentanoic	W	3.55 ± 0.45 ^Aa^	7.20 ± 0.97 ^Bb^	22.56 ± 3.07 ^Bd^	29.43 ± 1.89 ^Be^	12.66 ± 0.65 ^Bc^	10.28 ± 2.18 ^Bc^
		R	5.46 ± 0.24 ^Aa^	18.68 ± 1.04 ^Ab^	54.22 ± 3.43 ^Ad^	72.45 ± 2.76 ^Ae^	51.36 ± 3.68 ^Ac^	55.46 ± 4.89 ^Ac^
974	Hexanoic acid	W	41.03 ± 5.59 ^Aa^	69.03 ± 6.06 ^Aa^	73.39 ± 4.26 ^Bc^	83.76 ± 17.58 ^Ad^	63.92 ± 5.53 ^Bb^	58.76 ± 11.86 ^Bbc^
		R	2.76 ± 0.18 ^Ba^	13.23 ± 4.91 ^Ba^	90.30 ± 23.86 ^Ad^	75.86 ± 4.52 ^Bc^	73.01 ± 3.59 ^Ab^	70.74 ± 1.88 ^Ab^
1017	Heptanoic acid	W	4.89 ± 0.96 ^Ac^	3.42 ± 0.54 ^Ab^	3.02 ± 0.19 ^Bab^	ND	7.25 ± 1.02 ^Ad^	1.98 ± 0.52 ^Ba^
		R	0.68 ± 0.18 ^Ba^	2.37 ± 0.53 ^Ab^	11.35 ± 2.79 ^Ad^	2.84 ± 0.39 ^b^	6.51 ± 0.79 ^Ac^	10.01 ± 0.05 ^Ad^
1272	Nonanoic acid	W	1.82 ± 0.03 ^a^	4.39 ± 0.25 ^c^	ND	ND	3.16 ± 0.75 ^Ab^	2.47 ± 0.73 ^Bab^
		R	ND	ND	0.37 ± 0.16 ^a^	1.06 ± 0.09 ^c^	2.93 ± 0.26 ^Ac^	5.79 ± 0.68 ^Ad^
1372	Decanoic acid	W	4.62 ± 0.33 ^Ba^	18.59 ± 5.74 ^Ab^	25.13 ± 2.36 ^Ac^	28.04 ± 3.74 ^Ac^	25.40 ± 1.60 ^Ac^	20.93 ± 0.35 ^Ab^
		R	8.88 ± 0.72 ^Aa^	11.78 ± 0.59 ^Bb^	18.28 ± 0.52 ^Ba^	19.26 ± 0.65 ^Ba^	17.30 ± 1.05 ^Ba^	11.83 ± 2.48 ^Bb^
1173	Octanoic acid	W	32.15 ± 1.57 ^Aa^	56.31 ± 1.82 ^Ab^	91.57 ± 4.33 ^Ac^	75.13 ± 15.29 ^Ac^	65.43 ± 3.39 ^Ab^	56.86 ± 3.48 ^Ab^
		R	17.29 ± 1.98 ^Ba^	27.39 ± 3.08 ^Bbc^	43.76 ± 9.56 ^Bf^	37.24 ± 1.86 ^Be^	33.08 ± 15.52 ^Bcd^	24.72 ± 2.20 ^Bab^
1370	Dodecanoic acid	W	4.72 ± 0.24 ^Ad^	ND	0.60 ± 0.12 ^Aa^	1.43 ± 0.73 ^Ac^	1.91 ± 0.34 ^Bc^	2.11 ± 0.28 ^Bbc^
		R	0.61 ± 0.06 ^Ba^	ND	1.19 ± 0.16 ^Aa^	2.84 ± 0.22 ^Ab^	7.62 ± 1.49 ^Ac^	12.56 ± 1.56 ^Ad^
1249	Benzeneacetic acid	W	ND	0.80 ± 0.19 ^a^	2.59 ± 0.19 ^b^	ND	ND	ND
		R	ND	ND	ND	ND	ND	ND
1366	Undecanoic acid	W	ND	ND	ND	ND	ND	ND
		R	ND	ND	ND	ND	2.94 ± 1.05 ^a^	7.62 ± 0.18 ^b^
	Ketones							
594	2-butanone	W	10.80 ± 1.75 ^Bb^	23.44 ± 2.06 ^B c^	6.76 ± 0.95 ^Ba^	ND	ND	ND
		R	32.52 ± 1.04 ^Ac^	34.37 ± 3.21 ^Ac^	20.24 ± 1.77 ^Ab^	5.96 ± 0.35 ^a^	ND	ND
694	2-Pentanone	W	5.09 ± 0.35 ^Bb^	22.96 ± 0.55 ^Aa^	16.50 ± 1.23 ^Ac^	ND	ND	ND
		R	9.75 ± 0.55 ^Ab^	20.36 ± 1.93 ^Ad^	14.05 ± 0.58 ^Ac^	4.86 ± 0.73 ^a^	ND	ND
853	2-Hexanone	W	17.08 ± 3.79 ^Aa^	29.58 ± 3.29 ^Aa^	38.43 ± 5.69 ^Ba^	50.04 ± 1.10 ^Aa^	80.11 ± 15.67 ^Ab^	83.04 ± 7.54 ^Ab^
		R	8.49 ± 0.29 ^Ba^	12.38 ± 1.92 ^Ba^	45.88 ± 9.56 ^Aa^	49.52 ± 3.55 ^Aa^	77.03 ± 45.82 ^Bb^	82.62 ± 44.27 ^Aa^
952	2-Octanone	W	ND	ND	ND	7.13 ± 0.17 ^Aa^	9.81 ± 3.45 ^Aa^	15.16 ± 6.13 ^Ab^
		R	ND	ND	ND	2.74 ± 1.81 ^Ba^	5.86 ± 0.67 ^Bb^	5.66 ± 1.04 ^Bb^
1052	2-Nonanone	W	58.89 ± 8.14 ^Aa^	49.02 ± 1.73 ^Ba^	64.38 ± 13.59 ^Ab^	69.90 ± 6.70 ^Bb^	99.30 ± 3.35 ^Ac^	81.33 ± 9.44 ^Ac^
		R	50.64 ± 3.39 ^Ba^	55.84 ± 3.17 ^Aa^	48.47 ± 19.62 ^Ba^	78.47 ± 8.59 ^Ab^	87.21 ± 13.78 ^Bbc^	75.08 ± 28.39 ^Ba^
1042	8-Nonen-2-one	W	ND	ND	ND	29.93 ± 1.34 ^Ba^	37.84 ± 4.57 ^Ab^	41.67 ± 5.70 ^Bc^
		R	ND	5.39 ± 0.89 ^a^	22.96 ± 1.18 ^d^	35.26 ± 1.74 ^Ab^	38.08 ± 2.46 ^Ac^	46.78 ± 0.62 ^Ac^
1151	2-Decanone	W	ND	ND	ND	4.63 ± 0.66 ^Ab^	3.93 ± 1.06 ^Aa^	7.15 ± 1.32 ^Ac^
		R	ND	ND	ND	1.08 ± 0.32 ^Ba^	1.93 ± 0.26 ^Ba^	2.97 ± 0.61 ^Bb^
1159	7-Decen-2-one	W	ND	ND	ND	1.94 ± 0.70 ^Aa^	4.82 ± 1.19 ^Ab^	9.38 ± 2.25 ^Ac^
		R	ND	ND	ND	2.33 ± 0.73 ^Ab^	1.56 ± 0.39 ^Ba^	3.80 ± 0.23 ^Bc^
1251	2-Undecanone	W	9.52 ± 2.31 ^Ab^	10.98 ± 0.35 ^Aa^	18.13 ± 0.96 ^Ab^	17.02 ± 1.14 ^Bb^	24.71 ± 2.28 ^Bb^	28.63 ± 2.05 ^Bc^
		R	9.02 ± 0.59 ^Aa^	11.84 ± 1.33 ^Ab^	17.98 ± 2.05 ^Ac^	29.28 ± 1.29 ^Ac^	32.15 ± 2.05 ^Ac^	56.45 ± 2.99 ^Ad^
1370	2-Tridecanone	W	ND	ND	ND	ND	ND	ND
		R	ND	ND	ND	5.79 ± 0.29 ^a^	12.26 ± 0.95 ^b^	35.67 ± 0.77 ^c^
	Alcohols							
663	Ethanol	W	3.63 ± 0.42 ^b^	1.55 ± 0.17 ^a^	ND	ND	ND	ND
		R	ND	ND	ND	ND	ND	ND
743	2,3-Butanediol	W	13.4 ± 0.84 ^Ab^	18.26 ± 1.01 ^Bc^	6.76 ± 0.75 ^Ba^	5.35 ± 0.17 ^Aa^	ND	ND
		R	11.69 ± 1.89 ^Ab^	21.97 ± 1.28 ^Ac^	24.32 ± 2.97 ^Ac^	4.69 ± 0.75 ^Aa^	6.12 ± 0.64 ^a^	ND
697	3-methyl-1-Butanol	W	3.19 ± 0.92 ^ab^	14.09 ± 3.38 ^c^	31.59 ± 3.07 ^d^	14.58 ± 1.58 ^Ac^	4.61 ± 1.06 ^Ab^	ND
		R	ND	ND	ND	5.86 ± 0.27 ^Ba^	6.34 ± 0.04 ^Ab^	7.25 ± 1.28 ^b^
681	Pentanol	W	ND	ND	ND	ND	ND	ND
		R	1.80 ± 0.25 ^a^	4.71 ± 0.83 ^b^	19.49 ± 1.46 ^d^	10.42 ± 2.37 ^c^	6.72 ± 1.33 ^b^	3.93 ± 0.56 ^b^
815	5-methyl- 2-Hexanol	W	ND	4.60 ± 1.51 ^Aa^	14.98 ± 0.12 ^b^	ND	ND	ND
		R	ND	1.97 ± 0.56 ^Ba^	ND	ND	ND	ND
960	2-Heptanol	W	ND	ND	ND	ND	8.41 ± 0.98 ^Ba^	11.61 ± 1.43 ^Ab^
		R	2.32 ± 0.75 ^a^	3.89 ± 0.69 ^a^	11.14 ± 1.52 ^b^	16.23 ± 2.93 ^c^	26.55 ± 3.02 ^Ad^	8.46 ± 1.19 ^Bb^
979	2-Octanol	W	1.36 ± 0.59 ^a^	4.31 ± 0.32 ^b^	3.85 ± 0.62 ^b^	1.78 ± 0.18 ^a^	ND	ND
		R	ND	ND	ND	ND	4.71 ± 0.29 ^b^	3.26 ± 1.06 ^a^
969	1-Octen-3-ol	W	5.77 ± 0.25 ^a^	4.19 ± 1.10 ^Ba^	17.79 ± 2.75 ^Ad^	19.55 ± 1.87 ^Ae^	25.67 ± 1.56 ^Aa^	20.62 ± 0.87 ^Aa^
		R	ND	7.57 ± 0.48 ^Aa^	10.06 ± 1.42 ^Bb^	15.29 ± 6.41 ^Bc^	13.62 ± 2.42 ^Bd^	15.73 ± 1.73 ^Bc^
1078	2-Nonanol	W	ND	ND	3.24 ± 0.35 ^Ba^	6.85 ± 0.25 ^Ab^	10.44 ± 0.65 ^Ac^	17.02 ± 1.87 ^Ad^
		R	16.77 ± 3.15 ^b^	9.34 ± 0.34 ^a^	14.75 ± 1.02 ^Ab^	7.64 ± 0.26 ^Aa^	5.36 ± 0.74 ^Aa^	7.16 ± 1.21 ^Ba^
1136	Phenylethyl Alcohol	W	3.76 ± 0.59 ^Bc^	7.96 ± 0.87 ^Be^	5.44 ± 0.17 ^Bd^	3.66 ± 0.44 ^Bc^	2.87 ± 0.06 ^Bb^	1.08 ± 0.01 ^Ba^
		R	11.02 ± 1.48 ^Aa^	24.34 ± 2.57 ^Ac^	21.32 ± 1.58 ^Ac^	18.43 ± 6.31 ^Abc^	17.86 ± 1.06 ^Ab^	18.89 ± 2.33 ^Ab^
1178	Decanol	W	2.43 ± 0.21 ^Bb^	1.16 ± 0.63 ^Aa^	ND	ND	ND	ND
		R	4.55 ± 0.85 ^Ab^	1.91 ± 0.09 ^Aa^	ND	ND	ND	ND
	Esters							
1386	delta-Decanolide	W	2.05 ± 0.10 ^Aa^	2.17 ± 0.04 ^Ab^	1.96 ± 0.56 ^Aa^	1.79 ± 0.15 ^Bab^	2.48 ± 0.49 ^Bb^	1.87 ± 0.23 ^Ba^
		R	2.42 ± 0.39 ^Aa^	2.78 ± 0.74 ^Abc^	2.46 ± 0.28 ^Ab^	3.62 ± 1.02 ^Ac^	4.63 ± 0.74 ^Ad^	4.16 ± 0.61 ^Ad^
1602	delta-Dodecalactone	W	ND	1.68 ± 0.27 ^Ab^	2.39 ± 0.75 ^Ac^	2.44 ± 0.18 ^Ac^	1.82 ± 0.11 ^Bb^	1.18 ± 0.03 ^Ba^
		R	ND	1.37 ± 0.79 ^Aa^	2.01 ± 0.54 ^Ab^	2.77 ± 0.07 ^Ab^	3.94 ± 0.76 ^Ac^	3.76 ± 0.34 ^Ac^
984	Ethyl hexanoate	W	ND	1.75 ± 0.02 ^Bb^	1.95 ± 0.54 ^Bb^	2.67 ± 0.67 ^Bc^	1.34 ± 0.23 ^Ab^	0.98 ± 0.47 ^Ba^
		R	5.69 ± 1.08 ^b^	4.91 ± 0.39 ^Aab^	8.66 ± 0.57 ^Ac^	10.66 ± 3.15 ^Ad^	7.30 ± 1.75 ^Bc^	7.67 ± 0.98 ^Ac^
796	Ethyl butyrate	W	ND	0.78 ± 0.06 ^ab^	2.07 ± 0.74 ^bc^	7.68 ± 2.35 ^d^	9.04 ± 0.32 ^d^	6.82 ± 0.94 ^c^
		R	ND	ND	ND	ND	ND	ND
1183	Ethyl caprylate	W	ND	1.36 ± 0.43 ^Bb^	1.87 ± 0.18 ^Bc^	2.14 ± 0.49 ^Bc^	0.75 ± 0.27 ^Ba^	ND
		R	2.04 ± 0.34 ^a^	3.62 ± 0.45 ^Ab^	4.85 ± 0.70 ^Ab^	3.88 ± 0.66 ^Ab^	6.72 ± 0.15 ^Ad^	5.41 ± 0.88 ^c^
963	Methyl pentanoic	W	ND	0.67 ± 0.16 ^a^	2.42 ± 1.33 ^b^	2.86 ± 0.14 ^b^	6.61 ± 0.17 ^c^	8.59 ± 0.93 ^d^
		R	1.63 ± 0.46 ^b^	1.06 ± 0.03 ^a^	ND	ND	ND	ND
1381	Ethyl decanoate	W	ND	ND	1.46 ± 0.11 ^Bb^	0.84 ± 0.27 ^Ba^	ND	ND
		R	0.61 ± 0.06 ^a^	3.18 ± 0.79 ^b^	4.64 ± 0.15 ^Ac^	7.83 ± 0.31 ^Ad^	7.25 ± 0.85 ^d^	7.62 ± 1.09 ^d^
1083	Heptyl acetate	W	ND	ND	ND	1.52 ± 2.24 ^a^	2.65 ± 0.48 ^b^	5.88 ± 0.36 ^c^
		R	ND	ND	ND	ND	ND	ND
981	Hexyl formate	W	ND	ND	ND	ND	ND	ND
		R	ND	ND	1.09 ± 0.16 ^b^	0.89 ± 0.07 ^a^	ND	ND
1081	Methyl pentanoic	W	ND	ND	ND	ND	ND	ND
		R	1.83 ± 0.28 ^b^	0.69 ± 0.14 ^a^	ND	ND	ND	ND

Results of experiments are expressed as the mean ± SD for each experimental group (*n* = 3). The significance of differences in the same row not sharing a common superscript lowercase letter are significantly different (*p* < 0.05). Means ± SD with different superscript capital letters were significantly different among cheese samples within the column of the same ripening period (*p* < 0.05). ND: Not detected.

## Data Availability

Data is contained within the article.

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
