# Peer review of "Effects of Monascus on Proteolysis, Lipolysis, and Volatile Compounds of Camembert-Type Cheese during Ripening"

_foods, 2022, doi:10.3390/foods11111662_

Round 1
Reviewer 1 Report
The authors have submitted an interesting article on the shelf-life of Camembert-type cheeses produced using Monascus as an adjunct starter. However, some additional information is needed. Changes in the active acidity (pH) and microbiological quality of the cheese during storage are important parameters that should be monitored. These parameters should be included in the paper; I hope that they were determined in the experiment. Moreover, there is no sensory analysis of the product indicating whether cheese made using this additive would be acceptable to consumers. A sensory analysis would add to the value of the work and contribute to its commercial development. The ‘Statistical analysis’ section does not explain how the statistical differences between means for each parameter were determined. Under Table 1, there is no description of the significances contained in the table.
Author Response
Dear Editor and Reviewers:
Thank you for your letter and for the reviewers’ comments concerning our manuscript entitled “Effects of monascus on proteolysis, lipolysis, and volatile compounds of camembert-type cheese during ripening” (ID: foods 17723873). Those comments are all valuable and very helpful for revising and improving our paper, as well as the important guiding significance to our researches. We have studied comments carefully and have made correction which we hope meet with approval. The main corrections in the paper and the responds to the reviewer’s comments are as flowing:
Reviewer :
- The authors have submitted an interesting article on the shelf-life of Camembert-type cheeses produced using Monascus as an adjunct starter. However, some additional information is needed. Changes in the active acidity (pH) and microbiological quality of the cheese during storage are important parameters that should be monitored.
Reply: We examined the acidity values during the experiment and the data were not put in the article due to space limitations.
These parameters should be included in the paper; I hope that they were determined in the experiment. Moreover, there is no sensory analysis of the product indicating whether cheese made using this additive would be acceptable to consumers. A sensory analysis would add to the value of the work and contribute to its commercial development.
Reply: I agree with you that sensory analysis is crucial for food products, and we did sensory evaluation of both cheeses during the test, but due to space limitations, we only put data from gas chromatography.
The ‘Statistical analysis’ section does not explain how the statistical differences between means for each parameter were determined. Under Table 1, there is no description of the significances contained in the table.
Reply: Sorry, the table notes have been revised.
Reviewer 2 Report
Referee report
Manuscript: foods 17723873
This manuscript entitled “Effects of Monascus on proteolysis, lipolysis, and volatile compounds of Camembert-type cheese during ripening” aims to evaluate the effect of the use of Monascus, as a promissory adjunct starter of Camembert-type cheese, on the proteolysis, lipolysis, and volatile compounds during ripening for 40 days.
The subject falls within the general scope of the journal and is a new and original contribution, in an area where the scientific knowledge available is still scarce.
General comments:
The relevance of this study can be improved in the Introduction section: in fact, the authors refer “the application of Monascus as an adjunct starter “ increase “the acceptance of cheese by Chinese consumers“, it is a positive point of view which highlights the need for this study. However, the positive health benefits of this application, can be improved. Why the Monascus supplements are used in “treatment and prevention of hypertension as well as the reduction of cholesterol and the improvement of circulatory system”? What are the molecules or the active principle? The readers of a scientific article need to know "the whys". On the other hand, is this relevant? These positive effects of supplements are present in the fermented food products with Monascus? The development of these ideas and questions would improve the manuscript from a scientific point of view.
This study was supported on a one-vat-cheese production per type of sample (W and R). The milk is the same, but the cheese manufactured, especially under laboratory conditions (vats of 30 L), can significantly influence the final result. For this reason, there should always be production replicates. Ideally 3 vats (n=3), but minimum 2 vats x 2 samples per vat (n=4) and not only laboratory replicates, as described in the manuscript. This is a major weakness in a dairy science article. It is necessary to solve this weakness and more laboratory work is necessary.
Some remarks to improve the manuscript:
L45-47: As previously mentioned, health issues should be further discussed in order to highlight the relevance of the study.
L28-31: This statement have a serious error “Presently, Penicillium camembert, Penicillium roqueforti and Geotrichum candidum are the main fungi starters used in the production of Camembert cheese, and its can be purchased through commercial channels [1]”. In fact Penicillium roqueforti is not used in production of any Camembert cheese. Other important negative fact of this statement is that reference [1: Milesi et all, 2007] never refer anything about these species.
L29: Penicillium camembert, L60: Penicillium camemberti, L78: P. candidum – scientific rigor and uniformity is essential in the species nomenclature and use italic.
L51-55: For a user-friendly reading of a scientific article, it is absolutely necessary that the objectives are clearly exposed. It is possible to understand the objectives, but they need to be improved: “The aim of this study was (or were)...”
L128-130: it is suggested to include more information: for example, what was the statistical ANOVA post hoc test used for comparison of the average values?
L411-459: Some references, like [4] are difficult to find on a “Web of science library” and are not written in English. The use of these references should only be made if they are essential. It is not a rule but references with DOI should be valued.
Al the graphs are completely imperceptible, they are not acceptable even for peer review evaluation.
Table 1: is not acceptable even for peer review reading. 1) There are compounds that never appear during ripening. Why these compounds are included in the manuscript? They are important? 2) There is no care in the presentation, for example, there are letters of the ANOVA analysis (a, b, c) unformatted, or located in the inferior line. 3) It is not well understood how was conducted the one-way analysis of variance (ANOVA)? Between samples (R and W), in the same samples during ripening time? The authors must clarify it at the bottom note of the table.
Author Response
Dear Editor and Reviewers:
Thank you for your letter and for the reviewers’ comments concerning our manuscript entitled “Effects of monascus on proteolysis, lipolysis, and volatile compounds of camembert-type cheese during ripening” (ID: foods 17723873). Those comments are all valuable and very helpful for revising and improving our paper, as well as the important guiding significance to our researches. We have studied comments carefully and have made correction which we hope meet with approval. The main corrections in the paper and the responds to the reviewer’s comments are as flowing:
Comments and reply:
- The relevance of this study can be improved in the Introduction section: in fact, the authors refer “the application of Monascus as an adjunct starter “increase “the acceptance of cheese by Chinese consumers“, it is a positive point of view which highlights the need for this study. However, the positive health benefits of this application, can be improved. Why the Monascus supplements are used in “treatment and prevention of hypertension as well as the reduction of cholesterol and the improvement of circulatory system”? What are the molecules or the active principle? The readers of a scientific article need to know "the whys". On the other hand, is this relevant? These positive effects of supplements are present in the fermented food products with Monascus? The development of these ideas and questions would improve the manuscript from a scientific point of view.
Reply—Dear reviewer, thank you for the comments to the submitted manuscript. We have revised the introduction and marked it in red.
- This study was supported on a one-vat-cheese production per type of sample (W and R). The milk is the same, but the cheese manufactured, especially under laboratory conditions (vats of 30 L), can significantly influence the final result. For this reason, there should always be production replicates. Ideally 3 vats (n=3), but minimum 2 vats x 2 samples per vat (n=4) and not only laboratory replicates, as described in the manuscript. This is a major weakness in a dairy science article. It is necessary to solve this weakness and more laboratory work is necessary.
Reply— Thank you. One vat can make many blocks of cheese, we analyzed with different blocks as repetition.
- L45-47: As previously mentioned, health issues should be further discussed in order to highlight the relevance of the study.
Reply—We have revised the introduction and marked it in red.
- L28-31: This statement have a serious error “Presently, Penicillium camembert, Penicillium roqueforti and Geotrichum candidum are the main fungi starters used in the production of Camembert cheese, and its can be purchased through commercial channels [1]”. In fact Penicillium roqueforti is not used in production of any Camembert cheese. Other important negative fact of this statement is that reference [1: Milesi et all, 2007] never refer anything about these species.
Reply—Thank you. We are sorry for such a mistake and we have revised the introduction and reference with marked them in red.
- L29: Penicillium camembert, L60: Penicillium camemberti, L78: P. candidum – scientific rigor and uniformity is essential in the species nomenclature and use italic.
Reply— We have revised the the manuscript.
- L51-55: For a user-friendly reading of a scientific article, it is absolutely necessary that the objectives are clearly exposed. It is possible to understand the objectives, but they need to be improved: “The aim of this study was (or were)...”
Reply—Thank you very much for your suggestion. We have revised the introduction and marked it in red.
- L128-130: it is suggested to include more information: for example, what was the statistical ANOVA post hoc test used for comparison of the average values?
Reply—Thank you sir. We have revised the manuscript.
- L411-459: Some references, like [4] are difficult to find on a “Web of science library” and are not written in English. The use of these references should only be made if they are essential. It is not a rule but references with DOI should be valued.
Reply—We have been revised the references and marked it in red.
- Al the graphs are completely imperceptible, they are not acceptable even for peer review evaluation.
Reply— We have been replaced higher definition figures.
Table 1: is not acceptable even for peer review reading. 1) There are compounds that never appear during ripening. Why these compounds are included in the manuscript? They are important?
Reply— Some compounds that do not appear in Table 1 have low contents and are only detected in some cheeses, which are not important for cheese flavor, so they are just not listed in the table1.
2) There is no care in the presentation, for example, there are letters of the ANOVA analysis (a, b, c) unformatted, or located in the inferior line.
Reply—We have been revised the table 1.
3) It is not well understood how was conducted the one-way analysis of variance (ANOVA)? Between samples (R and W), in the same samples during ripening time? The authors must clarify it at the bottom note of the table.
Reply— We have clarified it at the bottom note of the table.

Reviewer 3 Report
The manuscript deals with the use of Monascus in Camembert-type cheese to study its proteolysis, lipolysis and VOCs during cheese ripening. The paper is generally well written, however some suggestions could be done to the structure of the paper to improve its presentation.
Specific comments:
- The Introduction must be expanded, it must contain more recent references, new research findings on the topic and a comparison must be made with those studies.
- Under section 2.5, please mention the instrument model of GC and MS systems used.
- The procedure of SPME extraction from the cheese is not clearly stated. A better explanation should be included.
- Some pertinent references advised to cite:
1. doi: 10.1007/s10068-018-0459-1
2. doi: 10.3389/fnut.2021.649611
3. doi: 10.1109/ISOEN.2017.7968861.
4. doi: 10.1007/s10068-016-0133-4
- The quality of all the figures must be enhanced. The figures and texts are completely unreadable. They need larger lettering to be legible.
- In Table 1, is the cheese type C or R?
- In my opinion, some reported molecules can not be defined as flavor compounds of the cheese and their presence could be due to occasional contamination of fiber or not reliable identification by NIST. Please check.
- The English language and grammatical errors, typos must be revised throughout the manuscript.
- The authors must present limitations and drawbacks of the proposed approach.
Author Response
Dear Editor and Reviewers:
Thank you for your letter and for the reviewers’ comments concerning our manuscript entitled “Effects of monascus on proteolysis, lipolysis, and volatile compounds of camembert-type cheese during ripening” (ID: foods 17723873). Those comments are all valuable and very helpful for revising and improving our paper, as well as the important guiding significance to our researches. We have studied comments carefully and have made correction which we hope meet with approval. The main corrections in the paper and the responds to the reviewer’s comments are as flowing:
Comments and reply:
The manuscript deals with the use of Monascus in Camembert-type cheese to study its proteolysis, lipolysis and VOCs during cheese ripening. The paper is generally well written, however some suggestions could be done to the structure of the paper to improve its presentation.
Specific comments:
- The Introduction must be expanded, it must contain more recent references, new research findings on the topic and a comparison must be made with those studies.
Reply—Good suggestions. We have revised the introduction and marked it in red.
- Under section 2.5, please mention the instrument model of GC and MS systems used.
Reply—We have added the instrument model in revisied manuscript.
- The procedure of SPME extraction from the cheese is not clearly stated. A better explanation should be included.
Reply—We have modified the SPME extraction method in revisied manuscript.
- Some pertinent references advised to cite:
- doi: 10.1007/s10068-018-0459-1
- doi: 10.3389/fnut.2021.649611
- doi: 10.1109/ISOEN.2017.7968861.
- doi: 10.1007/s10068-016-0133-4
Reply: Four suggested references have been cited in the article.
- The quality of all the figures must be enhanced. The figures and texts are completely unreadable. They need larger lettering to be legible.
Reply: Changes have been made to the clarity of the figures.
- In Table 1, is the cheese type C or R?
Reply: Sorry, the symbol is written wrong. In the table are the flavors of both W and R cheeses.
- In my opinion, some reported molecules can not be defined as flavor compounds of the cheese and their presence could be due to occasional contamination of fiber or not reliable identification by NIST. Please check.
Reply— Some compounds that do not appear in Table 1 have low contents and are only detected in some cheeses, which are not important for cheese flavor, so they are just not listed in the table 1.
- The English language and grammatical errors, typos must be revised throughout the manuscript.
Reply— The article has been corrected for English, grammar and typos.
- The authors must present limitations and drawbacks of the proposed approach.
Reply— Sorry, I don't understand meaning here. What do you mean by the limitations and drawbacks of the proposed approach?

Reviewer 4 Report
The paper under consideration intends to evaluate the effect of the use of Monascus on the main biochemical reactions and on the formation of volatile compounds that occur during the curing of Camembert type cheese. It is based on a comparative study between cheeses produced with and without Monascus in relation to the main compounds resulting from the most important reactions during the cheese curing process.
The results are well presented and discussed, although the figures are of very poor quality, preventing readers from seeing the details. They must be corrected.
Throughout the text, and mainly in the result presentation and discussion chapter, references should be introduced to justify some theoretical statements.
The conclusions are very limited, considering only some of the compounds that show differences between the control and experimental cheeses, not responding to one of the main challenges that the authors launch at the end of the introduction. The main advantages of using Monascus based on the results obtained must be clearly mentioned and evaluated against the objectives of the work.
These are the most relevant generic comments about the submitted paper. Detailed suggestions as below.
Specific suggestions:
Line 72 – 1.0x105 must be replaced by 1.0x105
Line 72 – 1.0x104 must be replaced by 1.0x104
Line 94 – withdraw “and-“
Lines 150-152 – introduce refª
Fig. 1 – Bad quality. Replace it with a good quality figure so that the details can be observed. It is suggested to use the same scale (X) in both graphs so that readers have a real idea of the comparative evolution between the two indices.
Fig. 2 - Bad quality. Replace it with a good quality figure so that the details can be observed.
Lines 196-198 - introduce refª
Fig 3 - Bad quality. Replace it with a good quality figure so that the details can be observed.
Line 213 – Substitute “As the ripening time” by “As the ripening time evolves”
Fig 4 - Bad quality. Replace it with a good quality figure so that the details can be observed.
Line 236 – English revision needed
Fig 5 - Bad quality. Replace it with a good quality figure so that the details can be observed.
Lines 243-246 - introduce refª
Lines 261-266 - introduce refª
Lines 278-279 - English revision needed
Lines 290-291 - introduce refª
Line 295 – substitute “by” by “from”
Lines 294-295 - introduce refª
Lines 302–306 - introduce references
Line 315 – substitute “pervious” by “previous”
Lines 323-325 - introduce refª
Lines 331-333 - introduce refª
Lines 335-336 - introduce refª
Lines 344-345 - introduce refª
Lines 347-350 - introduce references
Lines 359-360 - introduce references
Lines 366-373 - introduce references
Table 1 - Configure numbers in columns so that each result occupies only a single row
Conclusions – Needs a revision as suggested before
Author Response
Dear Editor and Reviewers:
Thank you for your letter and for the reviewers’ comments concerning our manuscript entitled “Effects of monascus on proteolysis, lipolysis, and volatile compounds of camembert-type cheese during ripening” (ID: foods 17723873). Those comments are all valuable and very helpful for revising and improving our paper, as well as the important guiding significance to our researches. We have studied comments carefully and have made correction which we hope meet with approval. The main corrections in the paper and the responds to the reviewer’s comments are as flowing:
Comments and reply:
- The paper under consideration intends to evaluate the effect of the use of Monascus on the main biochemical reactions and on the formation of volatile compounds that occur during the curing of Camembert type cheese. It is based on a comparative study between cheeses produced with and without Monascus in relation to the main compounds resulting from the most important reactions during the cheese curing process.
The results are well presented and discussed, although the figures are of very poor quality, preventing readers from seeing the details. They must be corrected.
Throughout the text, and mainly in the result presentation and discussion chapter, references should be introduced to justify some theoretical statements.
The conclusions are very limited, considering only some of the compounds that show differences between the control and experimental cheeses, not responding to one of the main challenges that the authors launch at the end of the introduction. The main advantages of using Monascus based on the results obtained must be clearly mentioned and evaluated against the objectives of the work.
Reply: Thanks to the reviewers for their excellent suggestions, we have revised the article according to them
These are the most relevant generic comments about the submitted paper. Detailed suggestions as below.
Specific suggestions:
Line 72 – 1.0x105 must be replaced by 1.0x105
Reply: This section has been revised
Line 72 – 1.0x104 must be replaced by 1.0x104
Reply: This section has been revised
Line 94 – withdraw “and-“
Reply: This sentence has been revised
Lines 150-152 – introduce refª
Reply: New literature has been cited in this section.
Fig. 1 – Bad quality. Replace it with a good quality figure so that the details can be observed. It is suggested to use the same scale (X) in both graphs so that readers have a real idea of the comparative evolution between the two indices.
Reply: Changes have been made to the figures in the article.
Fig. 2 - Bad quality. Replace it with a good quality figure so that the details can be observed.
Reply: Changes have been made to the figures in the article.
Lines 196-198 - introduce refª
Fig 3 - Bad quality. Replace it with a good quality figure so that the details can be observed.
Reply: Changes have been made to the figures in the article.
Line 213 – Substitute “As the ripening time” by “As the ripening time evolves”
Reply: Good advice, the sentence has been modified.
Fig 4 - Bad quality. Replace it with a good quality figure so that the details can be observed.
Reply: Changes have been made to the figures in the article.
Line 236 – English
Reply: A change has been made to the statement.
Fig 5 - Bad quality. Replace it with a good quality figure so that the details can be observed.
Reply: Changes have been made to the figure 5 in the article.
Lines 243-246 - introduce ref
Reply: Changes have been made to the figure 5 in the article.
Lines 261-266 - introduce refª
Reply: Based on the reviewers' suggestions, we have added new references
Lines 278-279 - English revision needed
Reply: We have made changes to the language.
Lines 290-291 - introduce refª
Reply: Based on the reviewers' suggestions, we have added new references
Line 295 – substitute “by” by “from”
Reply: Revised.
Lines 294-295 - introduce refª
Reply: Based on the reviewers' suggestions, we have added new references
Lines 302–306 - introduce references
Reply: Based on the reviewers' suggestions, we have added new references
Line 315 – substitute “pervious” by “previous”
Reply: Revised.
Lines 323-325 - introduce refª
Reply: Based on the reviewers' suggestions, we have added new references
Lines 331-333 - introduce refª
Reply: Based on the reviewers' suggestions, we have added new references
Lines 335-336 - introduce refª
Reply: Sorry, there is no relevant reference.
Lines 344-345 - introduce refª
Reply: Sorry, there is no relevant reference.
Lines 347-350 - introduce references
Reply: Based on the reviewers' suggestions, we have added new references
Lines 359-360 - introduce references
Reply: Based on the reviewers' suggestions, we have added new references
Lines 366-373 - introduce references
Reply: This is the result of our experiment, no relevant reference was found.
Table 1 - Configure numbers in columns so that each result occupies only a single row
Reply: Based on the reviewers' suggestions, we have added new references
Conclusions – Needs a revision as suggested before
Reply: We have partially modified the conclusions.

Round 2
Reviewer 1 Report
I renew my request for information on changes in acidity (pH) and microbiological quality of cheeses during storage.
Author Response
I renew my request for information on changes in acidity (pH) and microbiological quality of cheeses during storage.
Reply: Thank you for your advice. Due to the length of the article, microbiological and acidity value indicators are provided in the form of attachments.

Reviewer 2 Report
Referee report – (second round)
Manuscript: foods 17723873
This manuscript entitled “Effects of Monascus on proteolysis, lipolysis, and volatile compounds of Camembert-type cheese during ripening” aims to evaluate the effect of the use of Monascus, as a promissory adjunct starter of Camembert-type cheese, on the proteolysis, lipolysis, and volatile compounds during ripening for 40 days.
The subject falls within the general scope of the journal and is a new and original contribution, in an area where the scientific knowledge available is still scarce.
General comments:
The relevance of this study can be improved in the Introduction section: in fact, the authors refer “the application of Monascus as an adjunct starter “ increase “the acceptance of cheese by Chinese consumers“, it is a positive point of view which highlights the need for this study. However, the positive health benefits of this application, can be improved. Why the Monascus supplements are used in “treatment and prevention of hypertension as well as the reduction of cholesterol and the improvement of circulatory system”? What are the molecules or the active principle? The readers of a scientific article need to know "the whys". On the other hand, is this relevant? These positive effects of supplements are present in the fermented food products with Monascus? The development of these ideas and questions would improve the manuscript from a scientific point of view. (Solved)
This study was supported on a one-vat-cheese production per type of sample (W and R). The milk is the same, but the cheese manufactured, especially under laboratory conditions (vats of 30 L), can significantly influence the final result. For this reason, there should always be production replicates. Ideally 3 vats (n=3), but minimum 2 vats x 2 samples per vat (n=4) and not only laboratory replicates, as described in the manuscript. This is a major weakness in a dairy science article. Based only in this judgment, I don’t recommend the manuscript for publication in Foods (not in any 1st or 2nd quartile scientific journal). It is necessary to solve this weakness and more laboratory work is necessary. (This weakness wasn't solved, because need more laboratorial work. In my opinion, it is the main problem of this study. In scientific cheese studies without vat replicates we don´t know if the results are fortuity: the cheese manufacture, especially under laboratory conditions (vats of 30 L), can cause great variability in the product. The blocks cheese sampling can’t solve this weakness)
Some remarks to improve the manuscript:
L45-47: As previously mentioned, health issues should be further discussed in order to highlight the relevance of the study. (These ideas / questions were improved, but the introduction still can be improved)
L28-31: This statement have a serious error “Presently, Penicillium camembert, Penicillium roqueforti and Geotrichum candidum are the main fungi starters used in the production of Camembert cheese, and its can be purchased through commercial channels [1]”. In fact Penicillium roqueforti is not used in production of any Camembert cheese. Other important negative fact of this statement is that reference [1: Milesi et all, 2007] never refer anything about these Penicillium species). (Solved)
L29: Penicillium camembert, L60: Penicillium camemberti, L78: P. candidum – scientific rigor and uniformity is essential in the species nomenclature and use italic. (Solved)
L51-55: For a user-friendly reading of a scientific article, it is absolutely necessary that the objectives are clearly exposed. It is possible to understand the objectives, but they need to be improved: “The aim of this study was (or were)...” (Solved)
L128-130: it is suggested to include more information: for example, what was the statistical ANOVA post hoc test used for comparison of the average values? (Solved)
L411-459: Some references, like [4] are difficult to find on a “Web of science library” and are not written in English. The use of these references should only be made if they are essential. It is not a rule but references with DOI should be valued. (Solved)
Al the graphs are completely imperceptible, they are not acceptable even for peer review evaluation. (Solved)
Table 1: is not acceptable even for peer review reading. 1) There are compounds that never appear during ripening. Why these compounds are included in the manuscript? They are important? 2) There is no care in the presentation, for example, there are letters of the ANOVA analysis (a, b, c) unformatted, or located in the inferior line. 3) It is not well understood how was conducted the one-way analysis of variance (ANOVA)? Between samples (R and W), in the same samples during ripening time? The authors must clarify it at the bottom note of the table. (Partially solved. Being the aim of the study to compare two types of samples, W and R, why the authors only compare statistically samples along ripening? Why do not compare samples, W and R in the same age. It is possible to use different type of characters, for example a,b,c along ripening and A,B to compare R and W)
Author Response
Reply: Thanks for your good suggestions. Regarding the cheese replicates you mentioned in the review, in fact we have repeated the test several times and the error is basically acceptable. The introduction section has been further modified. We added a comparison of the flavor substances differences in two type cheeses at the same ripening time. Means ±SD with different superscript capital letters were significantly different among cheese samples within the column of the same ripening period (p < 0.05).

Reviewer 3 Report
The Introduction still needs more revision, adding some more relevant references.
I can not see the procedure of SPME extraction from the cheese stated. A better explanation should be included.
The suggested references have not been cited.
The authors must present the limitations and disadvantages of the study done.
Author Response
The Introduction still needs more revision, adding some more relevant references.
Reply: Thank you for your valuable suggestions. We have made further changes to the introduction section.
I can not see the procedure of SPME extraction from the cheese stated. A better explanation should be included.
Reply: Thank you for your suggestion, and we have made further changes for SPME extraction procedures in manuscript.
The suggested references have not been cited.
Reply: Suggested literature citations.
- doi: 10.1007/s10068-018-0459-1----reference 17
- doi: 10.3389/fnut.2021.649611----reference 18
- doi: 10.1109/ISOEN.2017.7968861. reference 32
- doi: 10.1007/s10068-016-0133-4---- reference 4
The authors must present the limitations and disadvantages of the study done.
Reply: Monascus has been used in food for thousands years, mainly for traditional foods such as fermented bean curd, red rice, red wine and red vinegar, etc. Monascus products are widely welcomed by Chinese consumers because of their natural red color and unique flavor. Many research results also showed that the fermentation products contain a variety of bioactive substances, such as monacarin K, γ-aminobutyric acid, red pigment and polysaccharide, etc. Therefore, Monascus are widely used as coloring agents, preservatives, nutritional supplements in food industry, and can also be used in the pharmaceutical industry as dietary supplements for improving blood pressure, lowering cholesterol levels, improving the circulatory system, etc. Therefore, the application of Monascus in cheese as a co-fermentative strain to improve the flavor and quality of Camembert-type cheese can not only enrich the variety of natural cheese but also improve the acceptance of cheese by Chinese consumers. In actual, Monascus has higher requirements for nutrition and environment, and the growth rate is lower than white mold, the future need to screen special Monascus strains suitable for cheese processing.
